# Coastal Forest Structure Survey and Associated Land Crab Population in Suao Dakenggu Community, Yilan, Taiwan

Chia-Hsuan Hsu [1,2], Wei-Ta Fang [3], Hung-Kai Chiu [4], Wei-Cheng Kao [5] and Tsung-Shun Huang [2,3,*]

1 Biodiversity Division, National Institute for Environmental Studies, Tsukuba 305-8506, Ibaraki, Japan
2 Taiwan Association for Marine Environmental Education, Taipei 112, Taiwan
3 Graduate Institute of Sustainability Management and Environmental Education, National Taiwan Normal University, Taipei 116, Taiwan
4 Dakenggu Community Development Association, Yilan City 270, Taiwan
5 Forestry Economics Division, Taiwan Forestry Research Institute, Taipei 100, Taiwan
* Correspondence: 80946002s@ntnu.edu.tw

**Abstract:** Coastal forests can increase the resilience of seaside communities against natural disasters. These forests also provide other benefits, including food and an avenue for economic growth. The Dakenggu community in Suao, Yilan (Taiwan), is adjacent to a coastal forest with an area of nearly 114,000 m². Artificial plantation has been performed locally in this area since 1977 to prevent the loss of beaches. The coastal forest area was estimated through drone aerial photography combined with a geographic information system. We found that *Pandanus tectorius* (11.5%), *Casuarina equisetifolia* (30.8%), *Cerbera manghas* (4.07%), *Hibiscus tiliaceus* (5.2%), and grass (23.52%) are the dominant species in the plant community of Dakenngu coastal forest, which together accounted for 75.1% of the total land area. The area covered by different species in the coastal forest was examined and estimated as well. The height and diameter at breast height (DBH) of the main tree species in five transects were surveyed, and we also found some significant differences among transects that correspond to cohorts planted at different times by the Forestry Bureau. We also performed a survey of land crabs in the same transects over five months to infer any differences in land crab species among the transects. We found that the transect dominated by *H. tiliaceus* had a larger population of land crabs than others. We revealed that the mudflat crab *Chiromantes haematocheir* prefers to live under *H. tiliaceus*. Finally, we propose recommendations for improving the biodiversity of the Dakenggu coastal forest so that it can become a sustainable resource for its residents.

**Keywords:** coastal forest; aerial photography survey; tally survey; land crab; Suao Dakenggu; Taiwan



## 1. Introduction

Mitigating the impact of natural disasters along coastlines is crucial. In the past, engineering methods, such as seawalls, levees, breakwaters, and dykes, have been applied to resist coastal waves. However, maintaining these artificial engineering facilities is costly; they may also have negative ecological impacts and increased socioeconomic costs [1]. Therefore, many studies focusing on green infrastructure have been conducted [2]. Green infrastructure refers to the use of interactive, complex ecosystems to construct facilities that are beneficial to society and the environment, as well as technological advancement [3]. Reguero et al. (2018) [4] believe that protecting the ecosystem of the coastline, such as coastal vegetation, coral reefs, dunes, and beaches, is also part of green infrastructure; maximizing the use of these ecosystems can be more cost-effective than constructing traditional engineering facilities. A coastal forest formed by the vegetation along a coast—a large area of green infrastructure—helps stabilize shorelines [5] and reduces shoreline erosion [6] and wave intensity [7,8]. Moreover, coastal forests can help areas adapt to the impact of sea-level rise [9–12]. It serves as the first protective barrier against natural disasters for people residing near seas. Coastal forests mainly slow down large waves that

accompany large natural disasters; for instance, coastal forests resist large waves caused by tsunamis [10,13] and storm surges [14,15]. Coastal forests are indispensable for local economies because they not only provide food [16,17], but are also tourism and recreation avenues [18]; they could also be a source of the under-forest economy and education in the future.

Dakenggu coastal forest is an artificial coastal forest without long-term monitoring data. According to the local residents, this forest was created by the Taiwanese government through afforestation over a large dune in this area approximately 30 years ago. This artificial coastal forest was created to prevent the coastline from retreating, the dunes from disappearing, and the northeast monsoon and typhoon from considerably affecting the area. Few coastal forest surveys have been conducted in this area. Only one publication has thus far reported on the vegetation in the Dakenggu coastal forest, but it mainly focused on seedling planting experiments on the north side [19]; it only indicated that pure *Casuarina equisetifolia* afforestation was performed in coastal forests in this district between 1977 and 1986. Coastal windbreaks cannot easily replace *C. equisetifolia*, which is a tree species that exhibits the tallest growth and greatest wind resistance compared to other coastal tree species. Because of the morphological characteristics of *C. equisetifolia*, including the high salt resistance of its cytoplasm, it has high drought and salt tolerance [20–24]. However, two pieces of documentation from 1977 and 1986 do not record information such as the area, scope, and quantity of planting [19]. Therefore, the current study is the largest survey of the Dakenggu coastal forest in Suao, Yilan, that discusses the successive afforestation or reforestation conditions over the past 40 years and investigates the composition of coastal forests and the gap (grass area) of coastal forests. Due to the lack of clear information on Dakenggu coastal forest, two of our aims in this study were (1) use a drone survey to understand the composition of the coastal forest; (2) use transects and tree tally to understand the differences in the diameter at breast height and height among each transect.

Land crabs' long-term monitoring is also important for the local people for developing environmental education programs and ecotourism. Since 2021, local communities, private companies, and scientists have been conducting land crab monitoring and restoration operations in this coastal forest [25]. Land crabs and coastal forests exhibit a close relationship, and the crabs can enhance the conditions and nutrient cycles of coastal forests [26–28]. For instance, *Cardisoma carnifex* has been observed to spread *Pandanus tectorius* seeds further [29]. Therefore, in this study, our third aim is to conduct a comprehensive survey of the coastal forest and use the data from past land crab surveys to understand the overall growth status of the coastal forest and the possible factors affecting the land crab population. In addition, this study provides suggestions for improving coastal forests in order to enhance their biodiversity.

## 2. Materials and Methods

Drones and combined tally field surveys on transects of coastal forests were used to explore the growth status of artificial coastal forests in Yilan. We chose the coastal forest in Suao, Yilan, adjacent to the Dakenggu community, as the research site, and we performed field surveys of the coastal forest along with a drone-based aerial photography survey. In addition, by using the past land crab survey data of this coastal forest, we determined the tree species habitat and distribution of land crabs.

### 2.1. Diameter at Breast Height and Tree Height Measurement

We established five coastal forest transects in July 2021, originally to investigate land crabs. Next, we conducted a survey of the diameter at breast height (DBH) and tree height in the five coastal forest transects on 19–20 March 2022, and the trees within 1 m on both sides of the five transects were investigated (Figure 1). The lengths of transect A, B, C, D, and F were 177.5, 150, 87.5, 96.1, and 188.5 (m), respectively. Because the Dakenggu coastal forest is a plantation forest, the four most abundant species are *P. tectorius*, *C. equisetifolia*, *Cerbera manghas*, and *Hibiscus tiliaceus*. DBH is the standard measure used to calculate a tree's

diameter; it is measured using a DBH ruler, with its unit being in centimetres. To measure tree heights, we used an ultrasonic altimeter (Vertex 5; Haglöf Sweden, Västernorrland County, Sweden) to measure the height of the highest trunk or branch of the trees in meters. We also compared the DBHs and tree heights of the same tree species in different transects and explored the reasons for any differences. For this, we used the normality test first to determine whether the date had a normal distribution; normally distributed data were analysed using one-way analysis of variance (ANOVA), whereas non-normally distributed data were first analysed using the Kruskal–Wallis test, followed by the Mann–Whitney U test for group differences. For the statistical analysis, we used IBM SPSS version 25.0.

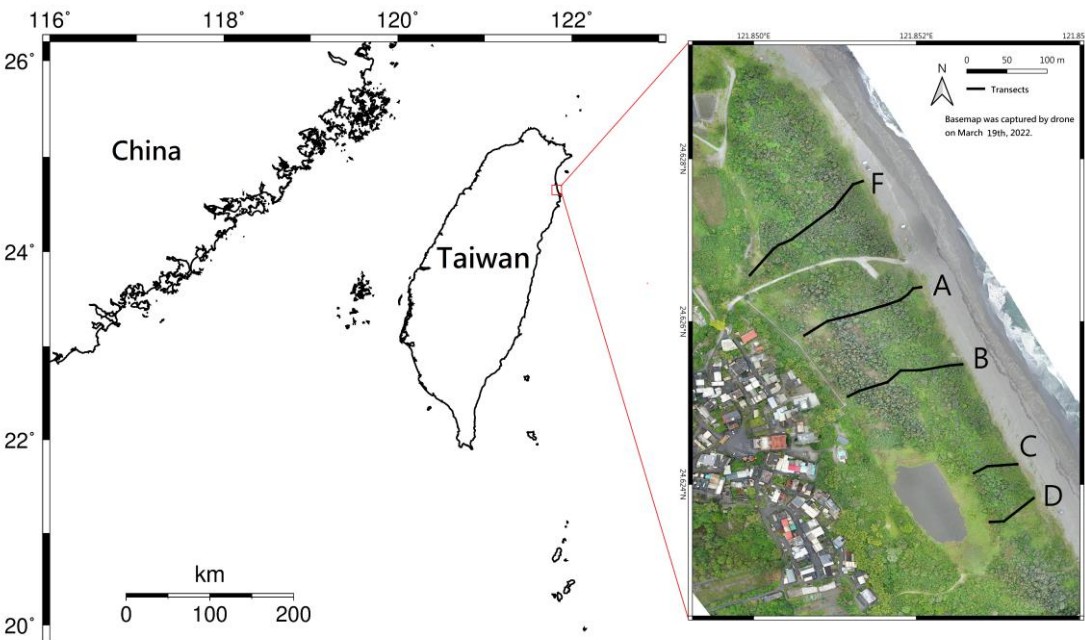

**Figure 1.** Research location and five transects considered for tally and land crab surveys in this study.

Next, we investigated whether the growth conditions of different tree species in the Dakenggu coastal forest were consistent among the transects. Therefore, we used regression analysis to determine the regression relationship and coefficient of determination ($R^2$) between DBH and tree height. A higher determination coefficient indicates more consistency.

### 2.2. Aerial Photography and Horizontal Distribution of Main Planting Tree Species

A drone (DJI Mavic 2 Pro) at a flight altitude of 60 m was used for the aerial photography of the coastal forest. The program DroneDeploy was used to automatically fly the drone and to obtain images. Next, we used Agisoft Metashape, a photo-editing software program, to analyse the composition of our aerial photographs and then used an AREY pro drawing tablet equipped with Photoshop 2021 to select and colorize tree species through manual identification: purple, brown, red, yellow, and blue were used to represent *P. tectorius*, *C. equisetifolia*, *Ce. manghas*, *H. tiliaceus*, and herbaceous voids, respectively. Subsequently, we used a geographic information system (GIS) (i.e., QGIS version 3.24.0) to analyse different areas. Polygon circle selection in QGIS's New Shapefile Layer function was used to draw the main planting tree species range, followed by the use of QGIS's Field Calculation function on the table properties to calculate the area of each tree species ($m^2$) using a coordinate system.

### 2.3. Land Crab Survey in the Coastal Forest

Surveys of land crab species were performed in the five transects once every month from July 2021 to November 2021. Approximately five people participated in each survey, and the species and numbers of land crabs in each transect were recorded. Data of the land

crab survey were stored at Dakenggu Community Development Association. This study combined the results of this land crab survey with the aforementioned vegetation survey results to explore the factors possibly affecting land crab species and population in the Dakenggu coastal forest. Finally, we used QGIS (version 3.24.0) to examine the relationship between the plant area and land crab population. Then, we used descriptive statistics and Fisher exact test in IBM SPSS to analyse the distribution of land crabs.

## 3. Results

### 3.1. Comparison of DBHs and Tree Heights of the Same Species in Different Transects

No *C. equisetifolia* data were collected from transect D because it has no *C. equisetifolia* trees. Therefore, only *C. equisetifolia* data from transects A, B, C, and F were included for further calculations.

We noted significant differences in tree heights between the different transects ($p < 0.001$, Kruskal–Wallis test; Table 1). Table 2 provides *C. equisetifolia* tree height percentile data for each transect. *C. equisetifolia* tree heights in transects A vs. B ($p < 0.001$, Mann–Whitney U test, A higher than B) and A vs. F differed significantly ($p < 0.001$, Mann–Whitney U test, A higher than F) (Table 1). *C. equisetifolia* in transect A were taller than those in transects B and F (Figure 2a). *C. equisetifolia* tree heights in the transects were in the following order: A = C ≥ F = B.

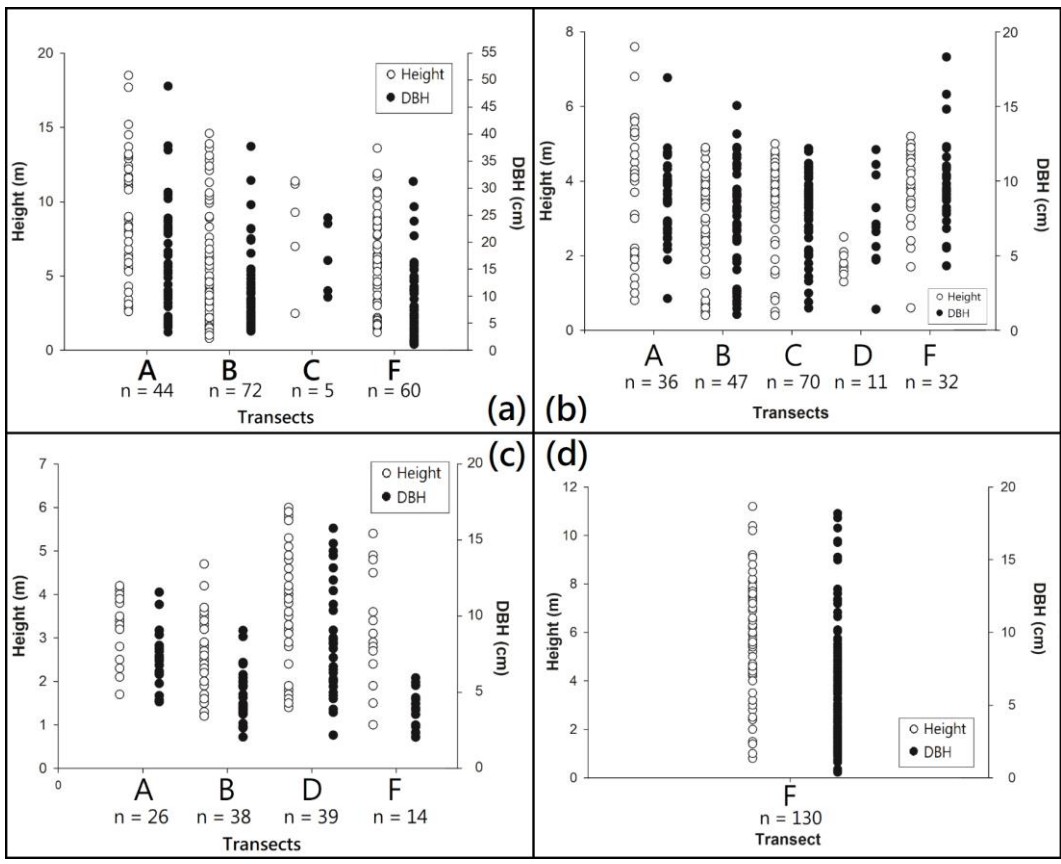

**Figure 2.** Comparison of tree height (white dots) and DBH (black dots) for the four main species in different transects: (**a**) *C. equisetifolia* (refer to Table 1 for significance); (**b**) *P. tectorius* (refer to Table 4 for significance); (**c**) *Ce. manghas* (refer to Table 7 for significance); (**d**) *H. tiliaceus*. The letters indicate the transects.

**Table 1.** Significance comparison of *C. equisetifolia* tree height and DBH in each transect (* *p* < 0.05, ** *p* < 0.01, *** *p* < 0.001). The letters indicate the transects.

| Transects | (V.S) | *U* Value | *p* Value of Tree Height | *U* Value | *p* Value of Tree DBH |
|---|---|---|---|---|---|
| A | B | 832 | <0.001 *** | 821 | <0.001 *** |
| A | C | 96.5 | 0.66 | 86 | 0.43 |
| A | F | 699.5 | <0.001 *** | 716 | <0.001 *** |
| B | C | 102 | 0.11 | 45 | 0.005 ** |
| B | F | 2098.5 | 0.78 | 2142 | 0.93 |
| C | F | 77.5 | 0.07 | 46 | 0.01 * |

**Table 2.** *C. equisetifolia* tree height in each transect (25th percentile, median, and 75th percentile, all in m). The letters indicate the transects.

| Transects | N | 25% | Median | 75% | Mean Rank |
|---|---|---|---|---|---|
| A | 44 | 5.6 | 8.9 | 12 | 122.5 |
| B | 72 | 2.3 | 4.5 | 7.4 | 78.6 |
| C | 5 | 5.9 | 9.3 | 11.3 | 118.4 |
| F | 60 | 2 | 5.7 | 8.2 | 80.5 |
| Total | 181 | 2.9 | 5.9 | 9.1 | |

The *C. equisetifolia* DBH differed significantly among the transects ($p < 0.001$, Kruskal–Wallis test; Table 1). Table 3 provides *C. equisetifolia* DBH percentile data for each transect. The difference in *C. equisetifolia* DBH between transects A and B ($p < 0.001$, Mann–Whitney U test, A bigger than B), A and F ($p < 0.001$, Mann–Whitney U test, A bigger than F), B and C ($p < 0.01$, Mann–Whitney U test, C bigger than B), and C and F ($p < 0.05$, Mann–Whitney U test, C bigger than F) were significant, but not among transects A and C and B and F. *C. equisetifolia* DBH in transect A was larger than that in transects B and F (Figure 2a). *C. equisetifolia* DBHs in the transects were in the following order: C = A > F = B on DBH.

**Table 3.** *C. equisetifolia* DBH in each transect (25th percentile, median, and 75th percentile, all in cm). The letters indicate the transects.

| Transects | N | 25% | Median | 75% | Mean Rank |
|---|---|---|---|---|---|
| A | 44 | 5.6 | 14.0 | 20.4 | 121.5 |
| B | 72 | 2.3 | 4.7 | 9.2 | 78.3 |
| C | 5 | 10.8 | 16.4 | 23.2 | 143.6 |
| F | 60 | 1.2 | 6.4 | 10.9 | 79.5 |
| Total | 181 | 2.6 | 6.8 | 13.2 | |

As is shown in Figure 3a, *C. equisetifolia* DBH versus tree height demonstrated a significant polynomial regression (F = 222, $p < 0.001$) with the following formula:

$$\text{Tree height} = -0.0081 \, (\text{DBH})^2 + 0.6606(\text{DBH}) + 1.5589, \, R^2 = 0.77$$

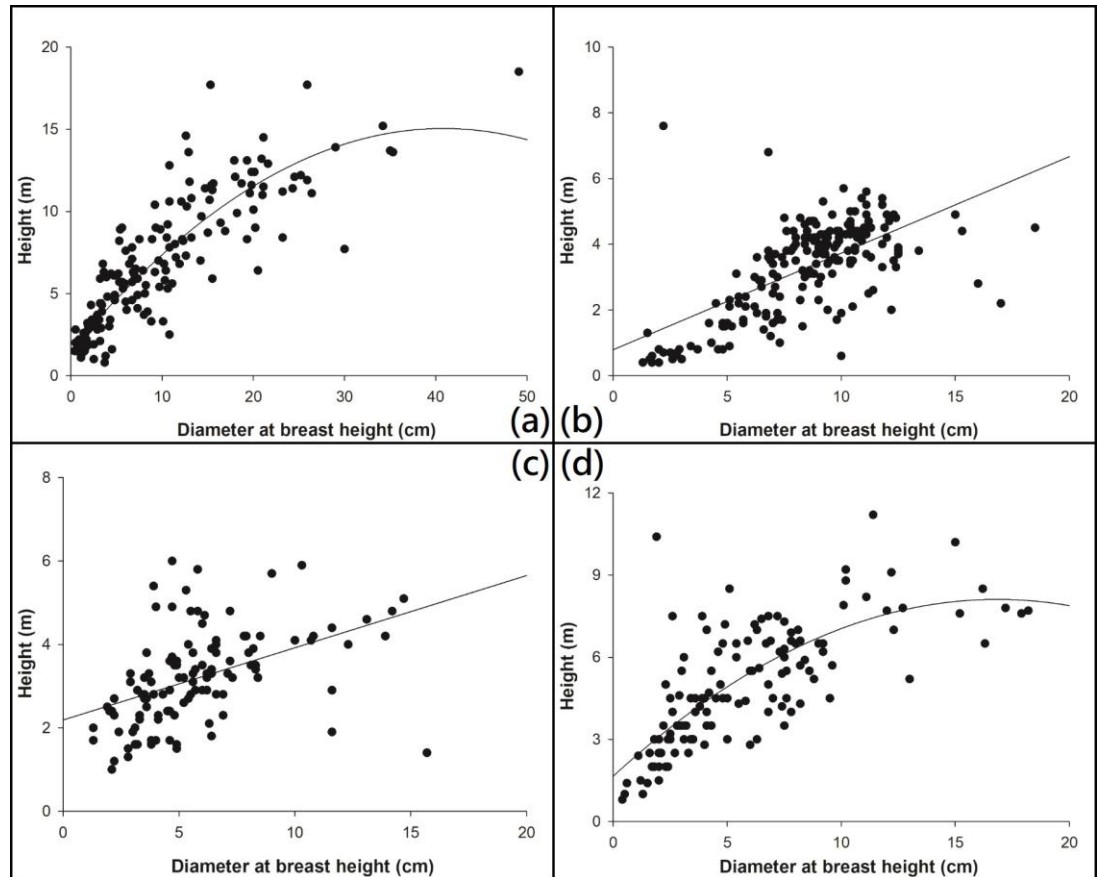

**Figure 3.** Regression analysis of DBH versus tree height. (**a**) Polynomial regression of *C. equisetifolia* ($p < 0.001$). (**b**) Linear regression of *P. tectorius* ($p < 0.001$). (**c**) Linear regression of *Ce. manghas* ($p < 0.001$). (**d**) Polynomial regression of *H. tiliaceus* ($p < 0.001$).

We noted significant differences in *P. tectorius* tree heights among the transects ($p < 0.001$, Kruskal–Wallis test; Table 4). Table 5 lists *P. tectorius* tree height percentile data for each transect. *P. tectorius* tree height was significantly different between transect A and B ($p < 0.05$, Mann–Whitney U test, A higher than B), A and D ($p < 0.01$, Mann–Whitney U test, A higher than D), B and C ($p < 0.05$, Mann–Whitney U test, C higher than B), B and D ($p < 0.05$, Mann–Whitney U test, B higher than D), B and F ($p < 0.01$, Mann–Whitney U test, F higher than B), C and D ($p < 0.001$, Mann–Whitney U test, C higher than D), and D and F ($p < 0.001$, Mann–Whitney U test, F higher than D) (Table 4). *P. tectorius* tree heights in the transects were in the following order: F = A = C > B > D (Figure 2b).

**Table 4.** Significance comparison of *P. tectorius* tree height and DBH in each transect (* $p < 0.05$, ** $p < 0.01$, *** $p < 0.001$). The letters indicate the transects.

| Transects | (V.S) | *U* Value | *p* Value of Tree Height | *U* Value | *p* Value of Tree DBH |
|:---:|:---:|:---:|:---:|:---:|:---:|
| **A** | **B** | 630 | 0.047 * | 660 | 0.09 |
| **A** | **C** | 1208.5 | 0.73 | 1109 | 0.32 |
| **A** | **D** | 68.5 | 0.001 ** | 138 | 0.13 |
| **A** | **F** | 550 | 0.75 | 420.5 | 0.06 |
| **B** | **C** | 1184 | 0.01 * | 1408 | 0.19 |
| **B** | **D** | 143.5 | 0.02 * | 242 | 0.74 |
| **B** | **F** | 491 | 0.009 ** | 442 | 0.002 ** |
| **C** | **D** | 98.5 | <0.001 *** | 300 | 0.24 |
| **C** | **F** | 1046 | 0.59 | 678.5 | 0.0014 ** |
| **D** | **F** | 17.5 | <0.001 *** | 93 | 0.02 * |

**Table 5.** *P. tectorius* tree height in each transect (25th percentile, median, and 75th percentile, all in m). The letters indicate the transects.

| Transects | N | 25% | Median | 75% | Mean Rank |
|:---:|:---:|:---:|:---:|:---:|:---:|
| A | 36 | 1.95 | 3.85 | 4.8 | 108.8 |
| B | 47 | 1.53 | 3 | 3.98 | 81 |
| C | 70 | 3.1 | 3.8 | 4.3 | 107.4 |
| D | 11 | 1.53 | 1.6 | 1.95 | 35.8 |
| F | 32 | 3.35 | 3.75 | 4.5 | 114.7 |
| Total | 196 | 2.15 | 3.6 | 4.3 | |

We also noted significant differences in *P. tectorius* DBH among the transects ($p < 0.01$, Kruskal–Wallis test; Table 4). Table 6 lists *P. tectorius* DBH percentile data for each transect. P. tectorius DBH differed significantly between transect B and F ($p < 0.01$, Mann–Whitney U test, F bigger than B), C and F ($p < 0.01$, Mann–Whitney U test, F bigger than C), and D and F ($p < 0.05$, Mann–Whitney U test, F bigger than D) (Table 4). *P. tectorius* DBHs in the transects were in the following order: F = A $\geq$ C = B = D (Figure 2b).

**Table 6.** *P. tectorius* DBH in each transect (25th percentile, median, and 75th percentile, all in cm). The letters indicate the transects.

| Transects | N | 25% | Median | 75% | Mean Rank |
|:---:|:---:|:---:|:---:|:---:|:---:|
| A | 36 | 7.1 | 8.9 | 10.3 | 105.2 |
| B | 47 | 4.9 | 7.4 | 9.7 | 83.3 |
| C | 70 | 7 | 8.6 | 9.8 | 94.6 |
| D | 11 | 5.1 | 7 | 10 | 76.3 |
| F | 32 | 8.7 | 10.2 | 11.2 | 129.4 |
| Total | 196 | 6.8 | 8.8 | 10.4 | |

As is shown in Figure 3b, *P. tectorius* DBH versus tree height demonstrated a significant linear regression ($p < 0.001$) with the following formula:

$$\text{Tree height} = 0.294\,(\text{DBH}) + 0.7883,\ R^2 = 0.39$$

In terms of *Ce. manghas*, there was no record of *Ce. manghas* in transect C. Significant differences were found in tree height in transects A, B, D, and F ($p < 0.05$, Kruskal–Wallis test; Table 7). Table 8 lists *Ce. manghas* tree height percentile data for each transect. *Ce. manghas* tree height was significantly different between transect A and B ($p < 0.05$, Mann–Whitney $U$ test, A higher than B) and B and D ($p < 0.05$, Mann–Whitney $U$ test, D higher than B). *Ce. manghas* tree heights in the transects were in the following order: D = A = F $\geq$ B (Figure 2c).

**Table 7.** Significance comparison of *Ce. manghas* tree height and DBH in each transect (* $p < 0.05$, ** $p < 0.01$, *** $p < 0.001$). The letters indicate the transects.

| Transects | (V.S) | U Value | p Value of Tree Height | U Value | p Value of Tree DBH |
|:---:|:---:|:---:|:---:|:---:|:---:|
| A | B | 234 | 0.0004 ** | 121 | <0.0001 *** |
| A | D | 424.5 | 0.27 | 426 | 0.28 |
| A | F | 162 | 0.57 | 48.5 | 0.0002 ** |
| B | D | 388.5 | 0.003 ** | 208.5 | <0.0001 *** |
| B | F | 194 | 0.14 | 230 | 0.46 |
| D | F | 220.5 | 0.29 | 90.5 | 0.0002 ** |

**Table 8.** *Ce. manghas* tree height in each transect (25th percentile, median, and 75th percentile, all in m). The letters indicate the transects.

| Transects | N | 25% | Median | 75% | Mean Rank |
|---|---|---|---|---|---|
| **A** | 26 | 2.8 | 3.5 | 3.9 | 66.6 |
| **B** | 38 | 2 | 2.7 | 3.2 | 41 |
| **D** | 39 | 2.8 | 3.6 | 4.8 | 71.5 |
| **F** | 14 | 2.4 | 3 | 4.5 | 59 |
| **Total** | 117 | 2.4 | 3.2 | 3.9 | |

We also noted significant differences in *Ce. manghas* DBH among the transects ($p < 0.001$, Kruskal–Wallis test; Table 7). Table 9 lists the *Ce. manghas* DBH percentile data for each transect. *Ce. manghas* DBH had significant differences between transect A and B ($p < 0.0001$, Mann–Whitney *U* test, A bigger than B), A and F ($p < 0.05$, Mann–Whitney *U* test, A bigger than F), B and D ($p < 0.0001$, Mann–Whitney *U* test, D bigger than B), and D and F ($p < 0.05$, Mann–Whitney *U* test, D bigger than F). *Ce. manghas* DBHs in the transects were in the following order: D = A > F = B (Figure 2c).

**Table 9.** *Ce. manghas* DBH in each transect (25th percentile, median, and 75th percentile, all in cm). The letters indicate the transects.

| Transects | N | 25% | Median | 75% | Mean Rank |
|---|---|---|---|---|---|
| **A** | 26 | 5.5 | 6.4 | 6.9 | 75.4 |
| **B** | 38 | 2.9 | 3.5 | 4.9 | 34.2 |
| **D** | 39 | 4.9 | 6.6 | 10.6 | 79.4 |
| **F** | 14 | 2.9 | 4.2 | 5.5 | 39 |
| **Total** | 117 | 3.7 | 5.3 | 6.7 | |

As is shown in Figure 3c, *Ce. manghas* DBH versus tree height demonstrated a significant linear regression ($p < 0.001$) with the following formula:

$$\text{Tree height} = 0.1733\,(\text{DBH}) + 2.187, \text{R}^2 = 0.21$$

*H. tiliaceus* was found only in transect F; therefore, the data could not be compared among the transects. Nevertheless, we analysed *H. tiliaceus* tree height and DBH data, including the distribution map in transect F (Figure 2d) as well as DBH and tree height percentiles (Table 10). As is shown in Figure 3d, *H. tiliaceus* DBH versus tree height demonstrated a significant polynomial regression (F = 76.9, $p < 0.001$) with the following formula:

$$\text{Tree height} = -0.0228\,(\text{DBH})^2 + 0.7669\,(\text{DBH}) + 1.6541, \text{R}^2 = 0.55$$

**Table 10.** *H. tiliaceus* tree height (m) and DBH (cm) in transect F (25th percentile, median, and 75th percentile). The letters indicate the transects.

| Transect | Category | N | 25% | Median | 75% |
|---|---|---|---|---|---|
| **F** | Tree height | 130 | 3.5 | 4.9 | 6.6 |
| **F** | Tree DBH | 130 | 2.9 | 5.1 | 7.8 |

### 3.2. Estimation of the Area of Coastal Forest Main Planting Tree Species

In this study, the Dakenggu coastal forest was divided into two areas based on vehicle access: the northern and southern coastal forests (Figure 4). The total area of the coastal forest investigated in this study was 113,676.14 m$^2$, with the areas of the north and the south coastal forests being 42,000.78 and 71,675.36 m$^2$, respectively (i.e., 36.9% and 63.1% of the total area, respectively).

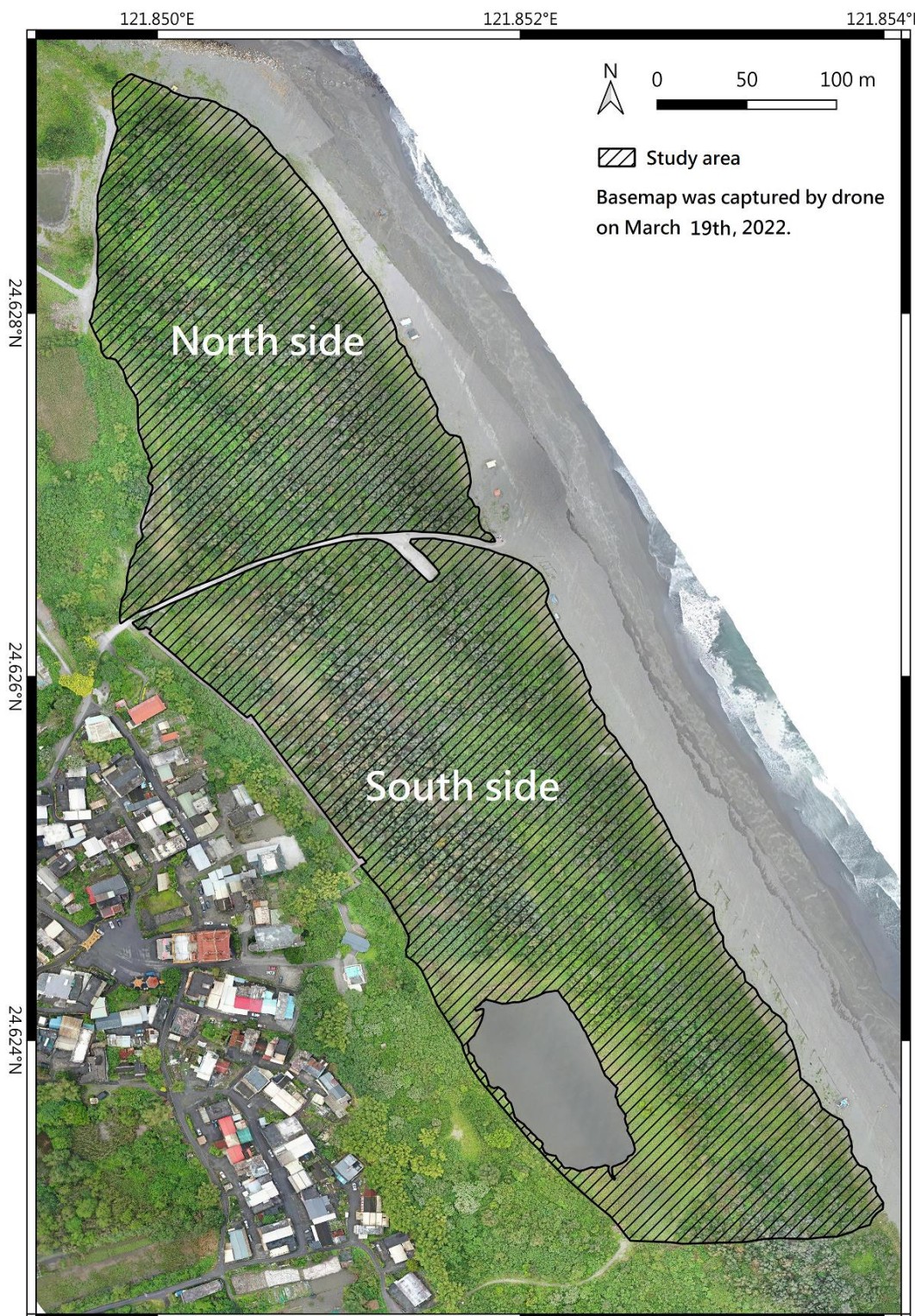

**Figure 4.** Coastal forest survey area (shaded with diagonal lines) considered in this study.

The four major plants and herbaceous voids (i.e., grass) investigated in this study accounted for 75.1% of the total coastal forest area. As is shown in Figure 5, we explored nearly all of the area as well as all species in this area. Table 11 presents the areas and proportions of all vegetation (main trees species and grasslands) surveyed in the total area as well as in its north and south sides.

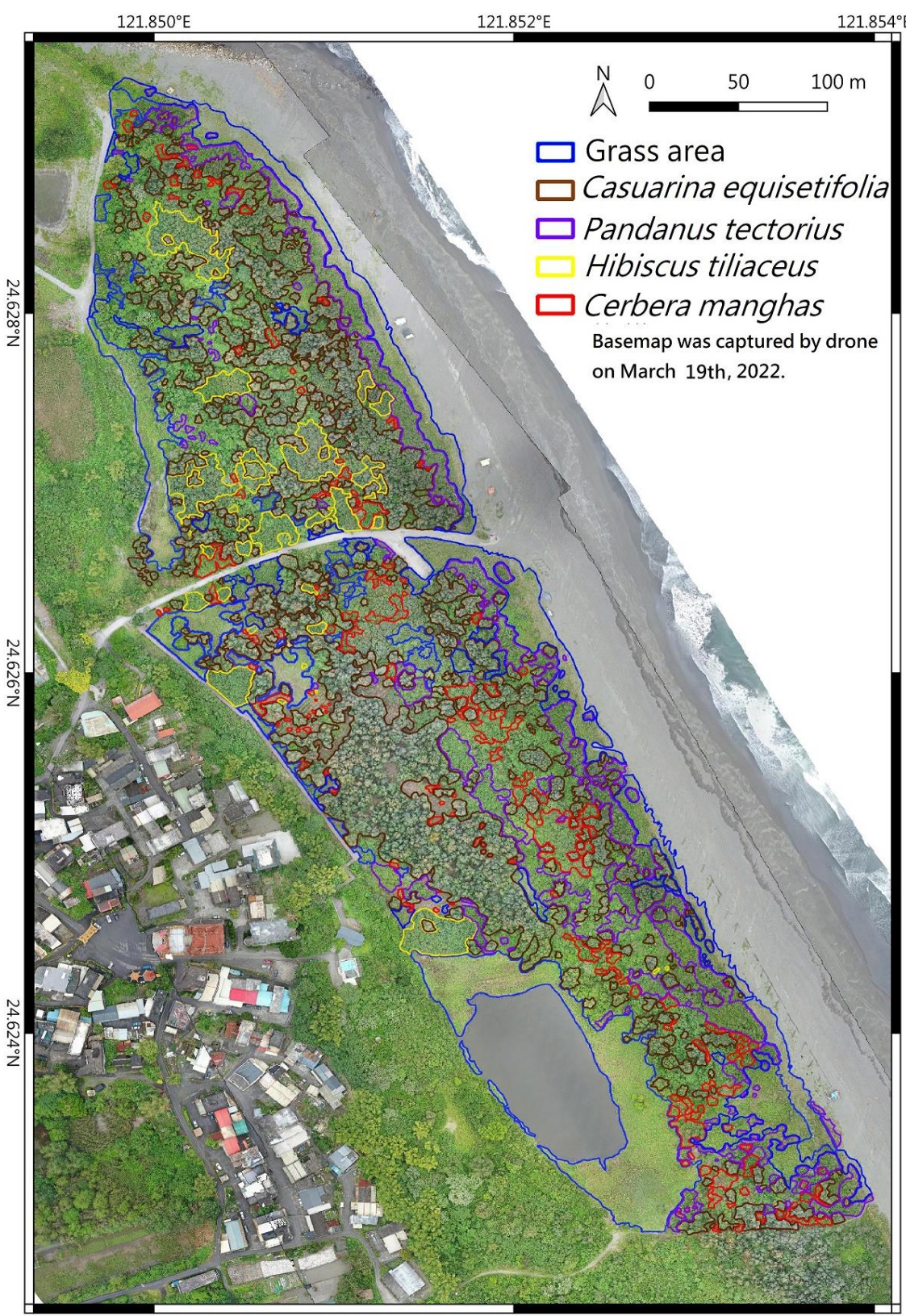

**Figure 5.** Main tree species and herbaceous voids (grass) surveyed in this study. Different colours represent the dominant tree species.

**Table 11.** Areas and proportions of all vegetation surveyed in this study.

| | North Side of Coastal Forest (m²) | Percentage of North Side (%) | South Side of Coastal Forest (m²) | Percentage of South Side (%) | Total Area of Each Species (m²) | Each Species in Total Percentage (%) |
|---|---|---|---|---|---|---|
| *Casuarina equisetifolia* | 13,598.5 | 32.38 | 21,410.85 | 29.87 | 35,009.35 | 30.8 |
| *Pandanus tectorius* | 3402.26 | 8.1 | 9670.53 | 13.49 | 13,072.79 | 11.5 |
| *Cerbera manghas* | 767.54 | 1.83 | 3857.78 | 5.38 | 4625.32 | 4.07 |
| *Hibiscus tiliaceus* | 4393.81 | 10.46 | 1520.28 | 2.12 | 5914.09 | 5.2 |
| Grass | 7407.04 | 17.64 | 19,334.07 | 26.97 | 26,741.10 | 23.52 |
| Others | 12,431.63 | 29.6 | 15,881.85 | 22.16 | 28,313.48 | 24.91 |
| Total | 42,000.78 | 100 | 71,675.36 | 100 | 113,676.14 | 100 |

*3.3. Land Crab Survey in the Coastal Forest*

The land crab survey was performed in the same transects as the vegetation survey (Figure 6); we noted that the main land crab species in the coastal forest were *Chiromantes haematocheir*, *Coenobita cavipes*, and *Metasesarma aubryi*. Of these, *Ch. haematocheir* was the most abundant in transect F. In total, 99 individuals were found over the five survey months, followed by seven in transect C. Moreover, we found six *M. aubryi* and nine *Co. cavipes* individuals in transect C. However, we did not observe many crabs in transects A, B, and D; no crabs were recorded in transect A over the five survey months. In other words, the number of individual land crabs in transects A, B, and D was deficient. Moreover, we should note that only transect F had *H. tiliaceus*. In contrast to transects B and F, transects A, C, and D have high percentages of grass coverage that are over 35% (Figure 6). Transect B had the highest coverage of *C. equisetifolia* at 52% (Figure 6).

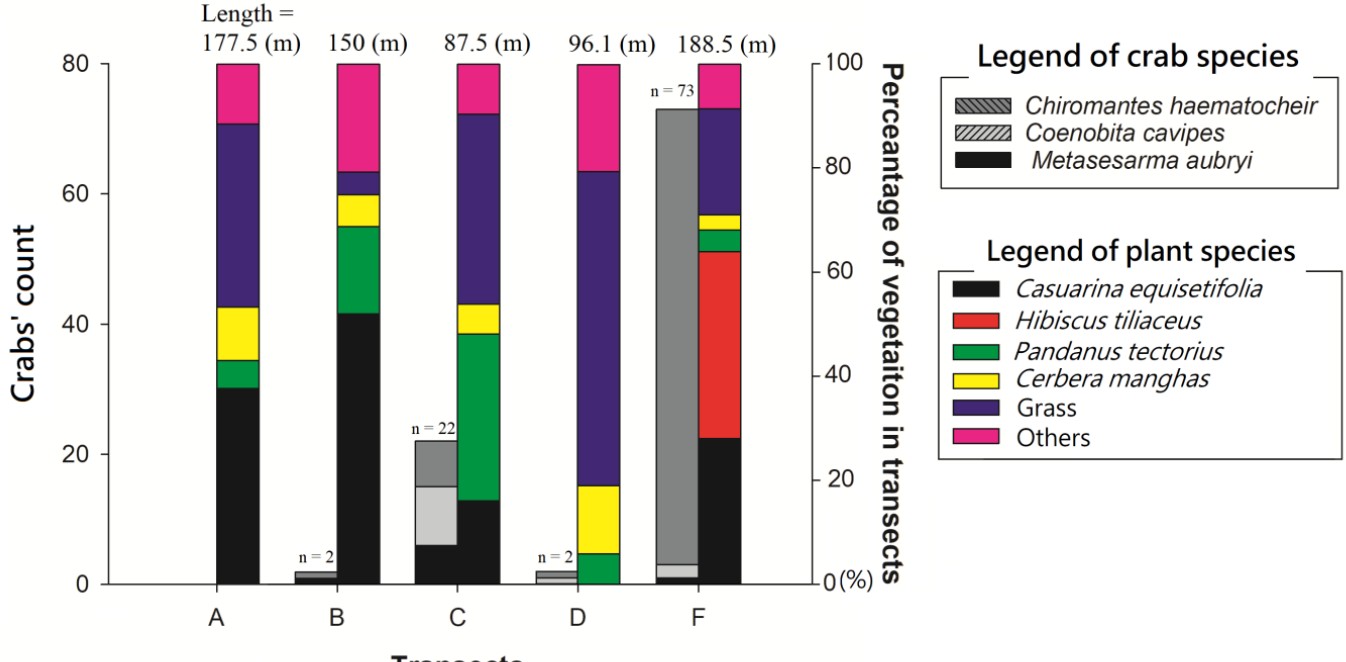

**Figure 6.** Results of the land crab survey and percentage of plant species for each transect in 5 months.

Figure 7 shows that most crab individuals (62.6%) were found under *H. tiliaceus*. A total of 65.8% of *Ch. haematocheir* individuals were found under *H. tiliaceus*, and 21.1% of individuals were found under *C. equisetifolia*. Regarding *Co. cavipes*, 50% of individuals were found under *P. tectorius*, and 41.7% were found under *H. tiliaceus*. Regarding *Metasesarma aubryi*, 62.5% of individuals were found under *H. tiliaceus*, and 37.5% of individuals were found under *P. tectorius*. The land crabs' distribution can be seen in Figure 8.

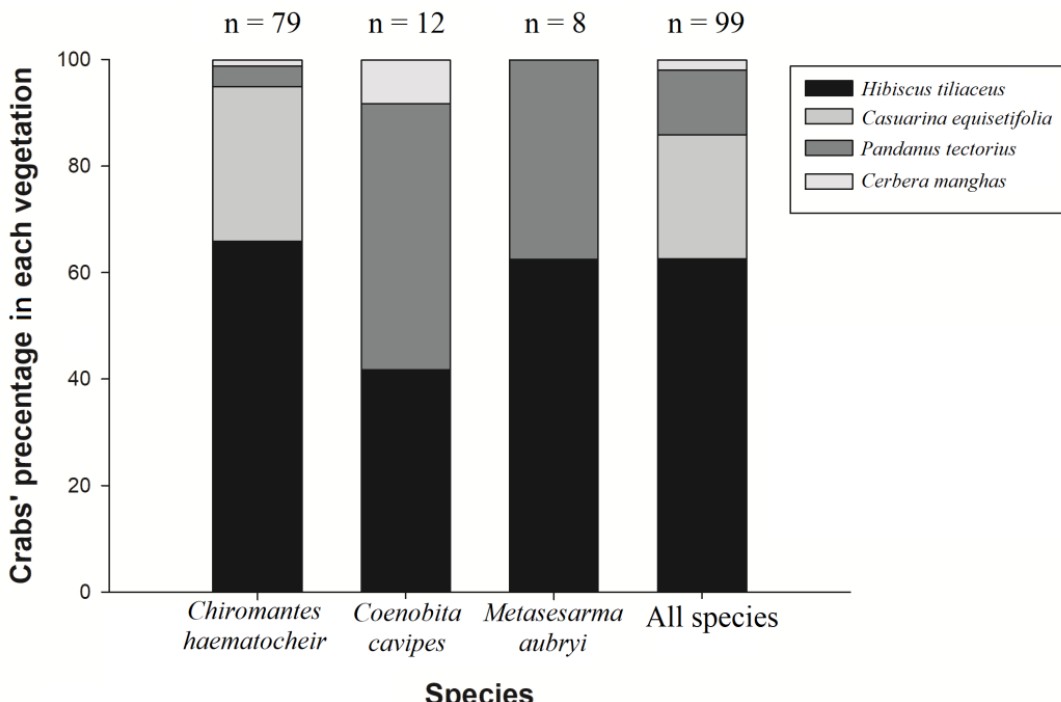

**Figure 7.** The result of crab species percentage in each vegetation species.

Because we found three species of crab under both *H. tiliaceus* and *P. tectorius*, we used Fisher's exact test to analyse whether the crabs showed habitat preference. Statistical analysis using Fisher's exact test (Table 12) indicated that the crabs' habitat showed significant differences. *Ch. haematocheir* was more associated with *H. tiliaceus* than *Co. cavipes* and *M. aubryi*. *Co. cavipes* and *M. aubryi* show no significant difference between the habitat with *H. tiliaceus* and *P. tectorius*.

**Table 12.** Differences between species under *Hibiscus tiliaceus* and *Pandanus tectoriu*. P = probability (Fisher exact test). *** $p < 0.001$, ** $p < 0.01$, * $p < 0.05$.

| | *Hibiscus tiliaceus* | *Pandanus tectoriu* | Fisher Exact Test *p* Value |
|---|---|---|---|
| *Chiromantes haematocheir* | 52 | 3 | |
| *Coenobita cavipes* | 5 | 6 | <0.001 *** |
| *Chiromantes haematocheir* | 52 | 3 | |
| *Metasesarma aubryi* | 5 | 3 | 0.02 * |
| *Coenobita cavipes* | 5 | 6 | |
| *Metasesarma aubryi* | 5 | 3 | 0.65 |

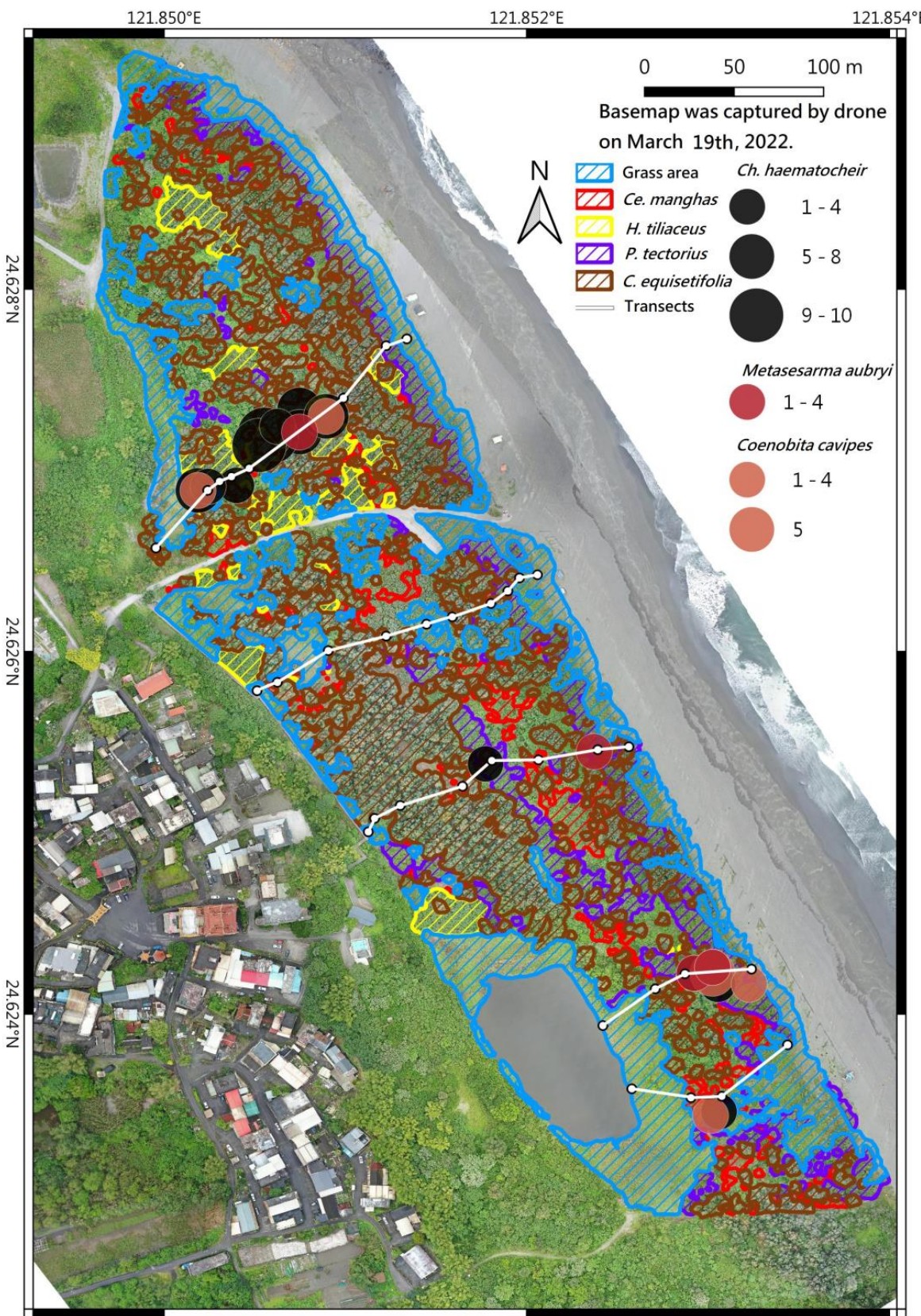

**Figure 8.** The land crabs' distribution over 5 months in Dakenggu coastal forest.

## 4. Discussion

*4.1. Comparison of Tree Heights of the Same Species with DBH in Coastal Forests*

According to previous reports, the coastal forests of Suao Dakkengu were planted in 1977 and 1986 [19]. However, not much information on the area, scope, and quantity was recorded [19]. Luodong Forest District Office (2010) documented that *C. equisetifolia*, *Millettia pinnata*, *H. tiliaceus*, *Melia azedarach*, and *Calophyllum inophyllum* reforestation was performed in 2010 on the north side of the study site [19]. However, according to this survey, other than *C. equisetifolia* and *H. tiliaceus* in the northern coastal forest, few tree species have been planted in this area. In addition, our results indicate that the tree heights and DBHs of the same tree species differ in different transects, corresponding to the traces of plantation in different stages and during different periods. The tree DBHs and heights may also be affected by different environmental factors, such as soil fertility and adjacent vegetation. In addition to *C. equisetifolia*, *P. tectorius*, *Ce. manghas*, and *H. tiliaceus* (mainly discussed in this study), we also recorded other tree species in the plantation forests, such as *Melaleuca leucadendra*, *Scaevola taccada*, and *M. pinnata*, in this survey. However, these species were not included in our analysis due to their rarity in our survey.

*C. equisetifolia* is an important tree species used for wind breaks, erosion control, and afforestation in tropical and subtropical regions [30]. In our study, we noted that *C. equisetifolia* was the youngest in transects B and F, with the 75th percentile of tree heights being only 7.4 and 8.2 m, respectively, which were lower than those of trees in transects A and C (12 and 11.5 m, respectively). During the investigation, we found many newly grown seedlings of *C. equisetifolia* in transects B and C. It has been found in previous studies that the seedling quality of *C. equisetifolia* may be influenced by fungus [31,32]. Thus, the different transects' microbial conditions might influence the structure of *C. equisetifolia*, and this requires further research. Inorganic fertilizers also influence the growth of *C. equisetifolia*. Nitrogen significantly influences seedling growth and biomass production; root length is significantly influenced by phosphorus, nitrogen–phosphorus, and nitrogen–phosphorus–potassium [33]. Only a few *C. equisetifolia* trees (n = 5) were noted in transect C, with large tree heights and DBHs; these may have been the remnants of the trees planted in the early years. We found that the DBH of *C. equisetifolia* demonstrated a polynomial regression relationship with tree height, with a high $R^2$ of 0.77, indicating that *C. equisetifolia* grows fairly uniformly in each transect and that tree height and DBH are not affected by other environmental factors such as soil fertility and adjacent vegetation. However, Liao et al. (2011) [34] reported no significant correlation between the DBH and tree height of *Casuarina*. In our study, *C. equisetifolia* trees on the north side have been planted since 2010 (for nearly 102 months) and have a tree height of 0.8–1.5 m. After 12 years, *C. equisetifolia* tree height at the 75th percentile in transect F on the north side was 8.2 m, which may be used as a basis for the growth height of 12-year-old *C. equisetifolia*.

*P. tectorius* trees had similar tree heights in transects A, C, and F; the 75th percentiles of tree height were 4.8, 4.3, and 4.5 m, respectively. Of these percentiles, *P. tectorius* trees in transect A were the largest, with a 75th percentile of 4.8 m for tree height; in general, the highest recorded *P. tectorius* tree height was 7.6 m in Dakenggu coastal forest. Our result for *P. tectorius* heights are similar to those of [35], which indicates that wild seedling-derived plants often have a single trunk for 4–8 m. In our results, we found that *P. tectorius* in Dakenggu coastal forest had similar DBHs but very different heights in each transect. The *P. tectorius* individual distribution was sparse in transect D, with only 11 trees being recorded, which were possibly newly planted. Notably, *P. tectorius* DBHs were similar in different transects, which indicates that *P. tectorius* grow DBH laterally first to occupy more space and then grow upward for sunlight exposure. This phenomenon was confirmed through our regression analysis, in which a linear regression relationship was found between *P. tectorius* DBH and tree height. However, here, the $R^2$ (=0.39) was lower than that for *C. equisetifolia*, which means that our regression model could explain only 39% of the predicted results. Nevertheless, studies have shown that the growth rate and

strategy of *P. odoratissimus*, another *Pandanus* species, exhibit differences across different environments [36].

We noted that transects A and D had similar *Ce. manghas* populations planted at the same time. Nevertheless, compared with transect A, transect D had *Ce. manghas* trees that were planted earlier, which had slightly larger tree heights and DBHs but were not significantly different. *Ce. manghas* in transects B and F were potentially planted during the same period and had relatively small DBHs and tree heights. Notably, the DBHs of *Ce. manghas* had significant differences between transects A and F and D and F; however, the tree heights did not differ significantly. We inferred that perhaps certain environmental factors influenced the *Ce. manghas* to grow higher in transect F. Although the *Ce. manghas* DBHs and tree heights revealed significant differences in the linear regression model, the $R^2$ value of 0.21 is very low. Thus, the growing model of *Ce. manghas* requires further research.

Because the area covered by *H. tiliaceus* in the southern coastal forest was small and fragmented, no *H. tiliaceus* was recorded in transects A, B, C, and D. According to our survey results, transect F in the northern coastal forest demonstrated a large, continuous area with *H. tiliaceus*. Elevitch and Thomson [37] indicate that the height of *H. tiliaceus* is 3–10 m, which is consistent with our result. Because only transect F demonstrated the presence of *H. tiliaceus*, relevant comparisons among transects could not be made, and only the regression relationship between tree height and DBH was obtained.

*4.2. Discussion on Coastal Forest Area*

The north and south sides of the Dakenggu coastal forest demonstrated differences in the types of studied vegetation, except *C. equisetifolia*. *C. equisetifolia* was highly abundant in both the northern (32.4%) and southern (29.9%) forest areas. The overall coastal forest area covered by *C. equisetifolia* was 30.8%—which was higher than that covered by other plants, possibly due to past afforestation or reforestation with *C. equisetifolia* as the main tree species in coastal forests. Other tree species accounted for 30% of the area in the northern coastal forest area, whereas the grass-covered area accounted only for 17.6% of the coastal forest area in the northern coastal forest area, which was lower than the 27% found in the southern coastal forest area. Therefore, the northern coastal forest area had more diverse vegetation than the southern area. In some areas, the forest phase was gradually replaced by native plants. However, *P. tectorius* accounted for 11.5% of the total coastal forest area. It was mostly distributed in the area adjacent to the sea and in coastal forest transect C in the south. *Ce. manghas* accounted for 4.1% of the total coastal forest area, most of which was newly planted in recent years. Moreover, the tree height and DBH data indicated that *Ce. manghas* trees were planted in two phases. The proportion of *H. tiliaceus* was nearly three times larger in the north than in the south. The litterfall of *H. tiliaceus* is more prone to retaining moisture, which may be related to the distribution of land crab populations that is discussed below.

Although many studies have revealed the structure of coastal forests [38–40], their methods and tree compositions are very different. Zhu, Matsuzaki and Gonda [38] used optical stratification porosity to find that Japanese coastal black pine (*P. thunbergii Parl.*) forests had different thinning ratios since 1997. Burley, Harper and Lundholm [39] used a perpendicular transect survey to qualify environmental variables such as tree age and height, soil properties, and bryophyte, vascular plant, and lichen species composition to understand the relationships between the composition of coastal forest and environmental variables across coastal forest–barren ecotones. The result indicated that tree age and height significantly differed throughout the ecotone compared to both the forest and the barren areas, but the soil properties were not significantly different across the transition compared to either the forest or the barren areas [39]. Kim and Choi [40] investigated four coastal forests in Busan, and the results revealed that all the sites' dominant species were *Pinus thunbergii* in the canopy layer, but the composition of the understory layers was different. Additionally, the trees also had significantly different DBHs and heights in four coastal forests [40]. However, these studies' survey methods and research purposes

and the compositions of coastal forests were very different from those of our study. Our study provides the first detailed survey of coastal forests in this area, and we plan to conduct long-term monitoring to understand the environmental change in the future.

### 4.3. Land Crab and Coastal Forest

This study also explored the potential reason why the land crab population in the southern Dakenggu coastal forest was relatively small. Even after 5 months of investigation, we did not observe any land crabs in transect A. Furthermore, transects B and D demonstrated sporadic land crab occurrence. Because we did not find any crabs in the grass area, we believe that the high percentage of grass area (over 35%) in transects A, C, and D led to habitat fragmentation and hindered the spread of the land crab populations. Grass area accounted for 23.5% of the total coastal forest area in this study. Although approximately 13% of the grass was noted to grow in the forefront between coastal forests and sandy land, it is a strong pioneer species. However, the grass-covered areas accounted for approximately 10% of the coastal forest, which may be the main reason underlying the fragmentation of the coastal forest habitat. We also observed that the forest facies composed of *C. equisetifolia* tended to have drier coastal forest bottoms. In transect B, which had the highest *C. equisetifolia* coverage (over 50%), we think high *C. equisetifolia* coverage may have caused a low humidity understory, leading to the small population of crabs. Although we found some crabs (23% of total crabs) under *C. equisetifolia*, most of them were found in transect F. We think this is because the high coverage rate of *Hibiscus tiliaceus* in transect F was correlated with a high abundance of crabs and overflow to adjacent *C. equisetifolia*.

*Ch. haematocheir* was the most common land crab species in the Dakenggu area. In the past, studies by ourselves and others have found that *Ch. haematocheir* requires high environmental humidity, which may be the reason for the sparseness of *Ch. haematocheir* in the southern coastal forest [41]. In the northern coastal forest, larger areas of *H. tiliaceus*, which have an understory at the bottom and are thus more likely to retain water, were noted. Forest trees' fallen branches and leaves are an important source of organic matter in forest lands. Woodland soil contents increase through the gradual decomposition of fallen foliage [42]. We speculate that this is the reason that more *Ch. haematocheir* were observed in transect F. Transect F belonged to the northern coastal forest in this study, and it demonstrated a continuous, large area of *H. tiliaceus* forest, covering 3247.79 $m^2$—larger than that noted in the southern coastal forest (1520.28 $m^2$). Our results also showed that only transect F had high coverage of *H. tiliaceus*. We do not know why transect F had high coverage of *H. tiliaceus*, but we propose that artificial planting in this coastal forest led to the uneven distribution of *H. tiliaceus*. We also found that *Ch. haematocheir* had a preference for *H. tiliaceus* over *Co. cavipes* and *M. aubryi*, which may indicate that *Ch. haematocheir* needs more water to survive. Our result also showed that only transect F had high coverage of *H. tiliaceus*. Moreover, we found that *Ch. haematocheir* had a greater population distribution under *H. tiliaceus* than *Co. cavipes* and *M. aubryi*, which may indicate that *Ch. haematocheir* needs more water to survive. Although land crabs are sometimes observed as roadkill, we encourage the community to participate in the Taiwan Roadkill Observation Network citizen science project. By uploading data, we can gain a better understanding of the roadkill situation in the Dakkengu community area, and participants can also learn something new about roadkill [43–45].

A large *P. tectorius* forest was noted in transect C, where most *Co. cavipes* and *M. aubryi* were found. We infer that the *P. tectorius* fruit could be an important food source for these two land crabs; the entire *P. tectorius* forest is fairly intact and does not contain any weeds that may hinder the crabs' migration path. Therefore, the numbers of individuals of these two species of land crabs were relatively large. However, compared with other locations where land hermit crabs are abundant, such as Lanyu, Xiaoliuqiu, and Xiziwan [46], and Kenting and Dongsha [47,48], these numbers were much smaller in Dakkengu area. Continual efforts to recover the land crabs' population are warranted. Recently, Huang and Hsu [49] reported that crab-eating mongooses (*Herpestes urva formosanus*) consumed

land hermit crabs in the coastal forest at Kenting, Taiwan. Thus, land hermit crabs (or land crabs) are potential food sources for other predators in coastal forests.

On the basis of our results, we recommend habitat restoration for land crabs in coastal forests through the removal and replanting of tree species that serve as food source plants, such as Moraceae plants, in weedy parts of the Dakenggu coastal forest. Although coastal forests are artificial, this study found that many other tree species slowly spread to coastal forests. The flowers, fruits, and seeds of these trees can be a food source for other organisms, such as *Delonix regia*, *Morus australs*, *Sapium discolor*, *Pongamia pinnata*, *Terminalia catappa*, and *Tournefortia argentea*. However, the number of individuals of these species remains small, possibly because the bottom soil of the coastal forest is of poor quality. The litter of *C. equisetifolia* has been reported to suppress the germination and initial growth of native trees on the Ogasawara Islands [50]. Therefore, if we hope to maintain high biodiversity in coastal forests, we should reduce the coverage of *C. equisetifolia* and increase the coverage of other native tree species. We recommend that some deciduous plants, such as yellow bark and terminalia, should be replanted in coastal forests to accelerate the accumulation of organic matter in the bottom of the coastal forest.

In conclusion, this study reported a comprehensive survey of the Dakenggu coastal forest and discussed the land crab population survey data on the area. We briefly propose some suggestions to make the Dakenggu coastal forest an environmental resource rich in biodiversity, which is beneficial to the communities around it via the intervention of not only the residents of the Dakenggu community but also other relevant stakeholders. A community-based citizen science community has been established in Dakenggu [25], and we hope to change the residents' attitudes and intentions regarding the promotion of local conservation action via land crab surveys in the coastal forest [51].

**Author Contributions:** Conceptualization, C.-H.H. and T.-S.H.; methodology, C.-H.H., W.-C.K. and H.-K.C.; software, C.-H.H. and H.-K.C.; validation, W.-T.F., T.-S.H. and H.-K.C.; formal analysis, C.-H.H. and H.-K.C.; investigation, C.-H.H., W.-C.K. and H.-K.C.; resources, W.-T.F. and T.-S.H.; data curation, H.-K.C.; writing—original draft preparation, C.-H.H. and W.-C.K.; writing—review and editing, C.-H.H. and T.-S.H.; visualization, C.-H.H. and H.-K.C.; supervision, T.-S.H.; project administration, C.-H.H. and W.-T.F.; funding acquisition, C.-H.H. and T.-S.H. All authors have read and agreed to the published version of the manuscript.

**Funding:** This research was funded by Urban Intertidal and Eaton Phoenixtec MMPL Co., Ltd.

**Data Availability Statement:** The data were stored at Dakenggu Community Development Association and are available for further use.

**Acknowledgments:** We would like to thank Chin-Ti Lin for helping us to capture the drone basemap. We thank Urban Intertidal, Eaton Phoenixtec MMPL Co., Ltd., and Taiwan Association for Marine Environmental Education for supporting this research. We also appreciate all the residents of Dakenggu community for giving us great support during this research.

**Conflicts of Interest:** The authors declare no conflict of interest.

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
