# Peer review of "Coastal Forest Structure Survey and Associated Land Crab Population in Suao Dakenggu Community, Yilan, Taiwan"

_diversity, doi:10.3390/d15040515_

Round 1
Reviewer 1 Report
Tally and Drone Survey of Coastal Forest Structure and Associated Land Crab Ecology and Population in Suao Dakenggu Community, Yilan, Taiwan
This article exhibits the results from authors’ survey of a coastal plantation, which is useful of assessing coastal plantations in similar areas.
comments:
This article presents some useful information related to the differences in plants and land crabs among the transects of the study site. However, some important information is missing. If I understand correctly, this study site should be a typical coastal wetland. If this is a coastal wetland, water level and land elevation are important factors impacting the vegetation distribution and crab population. If this is a tidal wetland, the tidal height should be presented, even if the study site is only impacted by the highest tides. The tree species should also be impacted by the porewater salinity. I don’t think it would be difficult to have the information related to soil salinity and tidal height, in addition to the soil fertility, to better understand the differences in the forest structure and crabs among the transects.
Line 321-322, If authors can give some information related to soil salinity and water level height, it may be better to understand the differences in DBH and height among the transects.
L.361-367, similarly, same species occurs in transects A, D and F, but DBH and height are different. This may be related to the differences in soil salinity and water table level among transects, in addition to differences in stand age. Authors should check whether there are some differences in soil properties and hydrology among the transects.
Although I do not comment on the grammar of this article, I still suggest that the author should check following sentences:
Line 74 – 75, … the difference of diameter at breast heights and heights between each transects.
Line 190, … between these transects. This should be “between” or "among". There are five transects in this study, not two. The author did not specify which two transects.
Line 199, similar to line 190, between the transects.
Line 229 again.
Line 372, same
Superscript:
Line 132, each tree species (m2) by using the coordinate system. m2 or m2.
Line 183, Tree height = -0.0081 (DBH)2 + 0.6606(DBH) + 1.5589, R² = 0.77.
Reviewer 2 Report
The authors submitted an interesting manuscript dealing with the use of Tally and Drone for coastal forest study in Suao Dakenggu Community, Yilan, Taiwan. The topic is interesting especially for researchers working in coastal areas mainly dealing with current changes in climate system. However, the authors need to provide more detailed description of the methodology (data acquisition, data processing and analysis) adopted to conduct this research. They should also check English language.
Author Response
Dear editor and reviewer,
Thank you for the time to review our paper and for giving us this chance to improve our article.
Thank you for giving us positive feedback. Yes, we also believe our research is very important for local people to face the current changes in the climate system. We already check the Method and Material in our article. We believe our data acquisition, data processing, and analysis are suitable. Besides, we also examined the English language. Thank you for your comments.
Round 2
Reviewer 1 Report
no more comments
Author Response
Thanks for reviewer's comments.
Reviewer 2 Report
No further comments
Author Response
Thanks for the reviewer's comments.